# A Multiple Twin-Roller Casting Technique for Producing Metallic Glass and Metallic Glass Composite Strips

**DOI:** 10.3390/ma12233842

**Published:** 2019-11-21

**Authors:** Yi Wu, Long Zhang, Sen Chen, Wen Li, Haifeng Zhang

**Affiliations:** 1School of Materials Science and Engineering, Shenyang Ligong University, 6 Nanping Middle Road, Shenyang 110159, China; 15104041255@163.com; 2Shenyang National Laboratory for Materials Science, Institute of Metal Research, Chinese Academy of Sciences, 72 Wenhua Road, Shenyang 110016, China; schen14s@imr.ac.cn (S.C.); hfzhang@imr.ac.cn (H.Z.); 3School of Materials Science and Engineering, Northeastern University, Shenyang 110819, China

**Keywords:** metallic glass, metallic glass composites, metallic strips, twin-roller casting

## Abstract

To date it has not been possible to produce metallic glass strips with a thickness larger than 150 μm via single-roller melt spinning technique, and it remains challenging to produce thick metallic glass strips. In this work, a multiple twin-roller casting technique is proposed for producing thick metallic glass and metallic glass composite strips. A triple twin-roller casting device, as a specific case of the multiple twin-roller, was designed and manufactured. The triple twin-roller device possesses a high cooling rate and involves a long contact time between the melt and the strip, which makes it an efficient technique for producing metallic glass strips that avoids crystallization, although the solidification temperature ranges of metallic glasses are as wide as several hundred Kelvins. The two prepared metallic glass (MG) strips are in a fully amorphous state, and the MG strip shows excellent capacity of stored elastic energy under 3-point bending. Furthermore, the Ti-based metallic glass composite strip produced via the triple twin-roller casting exhibits a novel microstructure with much finer and more homogenously orientated β-Ti crystals, as compared with the microstructure of metallic glass composites produced by the common copper mold casting technique.

## 1. Introduction

Metallic glasses (MGs) and metallic glass composites (MGCs) containing in-situ formed crystalline phases are potential structural materials due to their excellent mechanical properties, including high strength, high hardness and high elastic limit [1,2,3,4,5]. The in-situ formed crystals in MGCs are elaborately introduced for improving the plasticity of MGs [6,7,8,9,10,11]. MG and MGC strips, as initial materials, are widely used for making products via the thermoplastic forming process in the super-cooled liquid region [12,13]. Additionally, the yield strength, *σ*_y_, and the elastic limit, *ε*_e_, of MGs are generally 2–3 times that of their crystalline counterparts [2], which causes the stored elastic energy of MGs to be much higher than the conventional crystalline materials. Therefore, MG and MGC strips are ideal elastic materials for springs and clock winders [1,12].

However, MG strip with a thickness larger than 150 μm cannot be produced via single-roller melt spinning technique [14,15,16], and an efficient technique is required for making MG and MGC thick strips. Twin-roller casting technique has been adopted to produce steel and non-ferrous alloy strips [17,18], and recently, this technique has been occasionally used to produce Fe-based [19], Zr-based [20,21,22] and Mg-based [23] MG strips. However, full or partial crystallization often occurs in the MG strips produced via single (one set of twin rollers) twin-roller casting [19,20,21,23]. For example, Lee et al. [21] produced MG strips of Zr_41.2_Ti_13.8_Cu_12.5_Ni_10.0_Be_22.5_ (Vit.1) via single twin-roller casting, but all the strips partially crystalized into intermetallic crystalline phases, although the critical cooling rate of Vit.1 is as low as ~1K/s. This is because the solidification temperature range of MG, i.e., from the liquidus, *T*_l_, to the glass transition, *T*_g_, is as high as ~400 K, which is much larger than that of the crystalline alloys. The solidification range of crystalline alloys, i.e., from *T*_l_ to the solidus, *T*_m_, is generally smaller than 50 K [14,15]. The wide solidification temperature range of MGs requires a much higher cooling rate and a longer contacting time with the rollers.

In this work, we designed and manufactured a triple twin-roller device, as a prototype of the proposed multiple twin-roller casting technique. The results show that triple twin-roller casting technique is capable of producing long and straight MG or MGC thick strips. The prepared MG strip exhibits a much higher capacity of stored elastic energy than the conventional spring steels. Additionally, due to the squeezing flow during the multiple twin-roller casting, the produced Ti-based MGC strip exhibits a microstructure of fine and homogeneously-oriented β-Ti crystals.

## 2. Materials and Methods

The designed triple twin-roller device is shown in Figure 1. It contains six rollers with a diameter of 60 mm and a length of 90 mm, i.e., a triple twin-roller, a specific case of multiple twin-roller. The rollers were made of Cu-25wt.%Zn brass alloy for a good combination of high thermal conductivity and high hardenss. The brass rollers rotate at the same speed driven by an electric motor. One set of three rollers is attached to the springs and they are tightly contacted with the other set of fixed rollers under the spring force, *F*_spring_. The triple twin-roller device works in a high-purity argon atmosphere inside a furnace. The alloy pieces are melted by the induction coil and an infrared radiation thermometer monitors the real temperature. As the temperature is ~100 K higher than *T*_l_ of the alloy, the melt is blown by the Ar flow into the triple twin-roller device and an MG or MGC strip is then produced.

Two MGs and one MGC were selected for strip-making. One MG is Zr_41.2_Ti_13.8_Cu_12.5_Ni_10_Be_22.5_ (at.%, Vit.1) with a *T*_l_ of ~990 K and a *T*_g_ of 625 K, which possess a high-glass forming ability (GFA) [24]. The other one MG is Zr_55_Cu_30_Al_10_Ni_5_ (at.%, denoted as ZrCuAlNi) with a *T*_l_ of ~1150 K and a *T*_g_ of 690 K, which is one of the Be-free MGs with high GFA [25]. The selected MGC is Ti_45.7_Zr_33_Cu_5.8_Ni_3_Be_12.5_ (at.%, denoted as ZT-M) with a *T*_l_ estimated to be ~1475 K and a *T*_g_ of 602 K [26], which contains in-situ formed β-Ti crystals [27]. Both of the two MGs and the MGC have been well studied in their as-cast bulk forms. The roller speed and the Ar blowing-rate were kept the same for all the strips, which is 185 r min^−1^ and 8 L min^−1^, respectively. While, the nozzle diameter was 1.0 mm for the Vit.1 strip and 1.5 mm for the ZrCuAlNi and ZT-M strips. The parameters used for making the MG/MGC strips are listed in Table 1. Samples were cut from the strips for characterizations by means of X-ray diffraction (XRD; Cu-Kα, Bruker D8, Germany) and transmission electron microscopy (TEM, FEI Tecnai F20, USA). The specimens for conducting TEM investigations were prepared by ion-milling at a Gatan 691 device (Gatan Inc., Pleasanton, CA, USA) with the cooling of liquid nitrogen. 3-point bending tests at room temperature were performed on an Instron 5582 (Instron, Norwood, MA, USA) universal testing system with a loading rate of 0.1 mm min^−1^.

## 3. Results and Discussion

The produced MG/MGC strips are shown in Figure 2a–c. The Vit.1 MG strip exhibits a very smooth surface with metallic luster, see Figure 2a. The width is larger than 12 mm and the average thickness is 600 ± 20 μm. The length is more than 300 mm, which is much larger than the circumference (188.5 mm) of the brass roller. Therefore, a drastic decrease in the cooling rate occurred during the formation of the final part of the strip, and the final part did not fully solidify. As a result, the final part of the strip forms a curly shape. The dimensions of the prepared strips are also summarized in Table 1 for a better readability. In Figure 2d, the XRD spectrum of the final part of the Vit.1 strip only displays diffraction humps. Since the final part is of the lowest cooling rate, the XRD result suggests the full amorphous nature of the Vit.1 strip produced by the triple twin-roller casting.

The microstructure of the prepared Vit.1 strip was further characterized in TEM, as shown in Figure 3. The Vit.1 strip exhibits a featureless microstructure (Figure 3a), and only a diffuse halo ring was recorded in its selected-area electron diffraction (SAED) pattern (inset in Figure 3a). The high-resolution TEM (HRTEM) image (Figure 3b) of the Vit.1 strip shows maze-like patterns, and no long-range order of atomic arrangement can be noticed. The corresponding fast-Fourier transform (FFT) image (inset in Figure 3b) only exhibits a diffuse ring, which confirms the fully glassy nature of the Vit.1 thick strip.

The Vit.1 thick strip, as an example case, was further selected to show the excellent capacity of stored elastic energy under 3-point bending. The nominal stress (*σ*) and nominal strain (*ε*) can be calculated from the recorded data of the loading force (*F*) and the displacement (*δ*), respectively [28]:(1)$$\sigma =\frac{3FS}{{2bt}^{2}}$$
(2)$$\varepsilon =\frac{6t\delta }{S^{2}}$$
where *S*, *b*, and *t* is the span length, the width, and the thickness, respectively, of the 3-point bending tested samples. The typical nominal stress-strain curve of the Vit.1 strip is shown in Figure 4. It is noteworthy that the upper and bottom surfaces bear the highest stress during testing, and the distribution of stress on the cross-section is schematically shown in the inset of Figure 4. The yield strength of the surficial regions, i.e., *σ*_max_, is ~1800 MPa, which is comparable to its yield strength obtained under uniaxial compression or tension. After yielding, the Vit.1 strip also underwent a small stage of plastic deformation and was fractured at a total strain of 4.1%. The average density of stored elastic energy, $\bar{U}$ can be calculated by:(3)$$\bar{U}=\frac{1}{btS}\int_{-t/2}^{t/2}UbSdt$$
where U is the locally stored elastic energy per unit volume:(4)$$U=\frac{\sigma \varepsilon }{2}=\frac{\sigma^{2}}{2E}$$
and its distribution on the cross-section is shown in the inset of Figure 4. E = 95 GP is the elastic modulus of Vit. 1 MG [24]. Then the average density of stored elastic energy, $\bar{U}$, is obtained:(5)$$\bar{U}=\frac{\sigma_{\max}^{2}}{6E}\approx 5.6\times 10^{6}\;J\;m^{-3}$$

This is actually a very high stored elastic energy under 3-point bending. In comparison, the currently widely used spring steel possesses a yield strength of ~1500 MPa, and an elastic modulus of ~200 GPa [29], which implies that its average stored elastic energy under bending is only 1.8 × 10^6^ J m^−3^. Therefore, the capacity of stored elastic energy of the current Vit.1 strip is three times that of the conventional spring steels. The excellent capacity of stored elastic energy makes thick MG strips promising in applications as spring materials.

The prepared ZrCuAlNi strip is also wide and straight, as shown in Figure 2b. The width of the ZrCuAlNi MG strip is ~26 mm, which is much larger than the Vit.1 strip due to the use of a larger nozzle diameter. The length and thickness is ~190 mm and 430 ± 30 μm, respectively, as listed in Table 1. Although the GFA of ZrCuAlNi is much lower than Vit.1 [24,25], no diffraction peaks from crystalline phases can be detected in its XRD result, suggesting a fully amorphous state of the ZrCuAlNi strip and a high cooling rate of the triple twin-roller casting.

The microstructure of the prepared ZrCuAlNi strip was further characterized in TEM, as shown in Figure 5. The ZrCuAlNi strip displays a featureless microstructure (Figure 5a), and only a diffuse halo ring can be observed in its SAED pattern (inset in Figure 5a). The HRTEM image (Figure 5b) shows maze-like patterns with a lack of long-range order as supported by the diffuse ring in its FFT image. The TEM results also confirm the glassy nature of the produced ZrCuAlNi strip.

The produced ZT-M MGC strip looks similar to the ZrCuAlNi strip, see Figure 2c. The length is ~170 mm. Because of a higher viscosity of the MGC melt, the ZT-M strip has a smaller width (~20 mm) and a larger thickness (460 ± 30 μm) than the ZrCuAlNi strip, despite using the same nozzle diameter. The dimension of the prepared ZT-M strip is also listed in Table 1. The XRD spectrum of the ZT-M strip shows a typical feature of Ti-based MGCs containing β-Ti, i.e., the diffraction peaks from a body-centered cubic phase are superimposed on the diffraction hump from a glassy phase, see Figure 2d.

The microstructure of the prepared ZT-M strip was also characterized in TEM, as shown in Figure 6. The ZT-M strip clearly exhibits a composite microstructure (Figure 6a) with β-Ti crystals embedded in the glassy matrix. The average diameter of β-Ti crystals is only ~250 nm, one magnitude smaller than the crystal size in the ZT-M rods produced by copper mold casting [7,8]. This also implies the high cooling rate of the triple twin-roller casting. The HRTEM image (Figure 6b) from the interfacial region shows that there are no other intermetallic phase forms. Additionally, β-Ti crystals are generally in dendritic shape in the as-cast Ti-based MGC rods produced via copper mold casting, where a dendrite is a crystal [8,30]. In comparison, β-Ti crystals are mainly in particulate shape in the current strip, see Figure 6a. Furthermore, the remnant dendrite in the MGC strip is not a crystal anymore. They are dendritic fragments with different orientations, as clearly revealed by the bright-field and dark-field TEM images in Figure 6c,d. This implies that the single-crystal dendrite in the as-cast samples changes into several β-Ti crystals, as a result of the deformation flow during the multiple twin-roller casting.

## 4. Conclusions

A triple twin-roller, as a specific case of the multiple twin-roller, was designed and manufactured. The triple twin-roller device possesses a high cooling rate and a long contacting time, which enables it to be an efficient technique for producing thicker MG and MGC strips with a wide solidification temperature range. The Zr_41.2_Ti_13.8_Cu_12.5_Ni_10_Be_22.5_ (Vit.1) and Zr_55_Cu_30_Al_10_Ni_5_ MG strips as well as the Ti_45.7_Zr_33_Cu_5.8_Ni_3_Be_12.5_ (ZT-M) MGC strip were produced via triple twin-roller casting technique. The capacity of stored elastic energy of the Vit.1 strip under 3-point bending is as high as 5.6 × 10^6^ J m^−3^, which is three times that of the conventional spring steels. Furthermore, it is revealed that the MGC strip produced via the multiple twin-roller casting exhibits a unique microstructure with much finer and more homogenously orientated β-Ti crystals.

## Figures and Tables

**Figure 1 materials-12-03842-f001:**
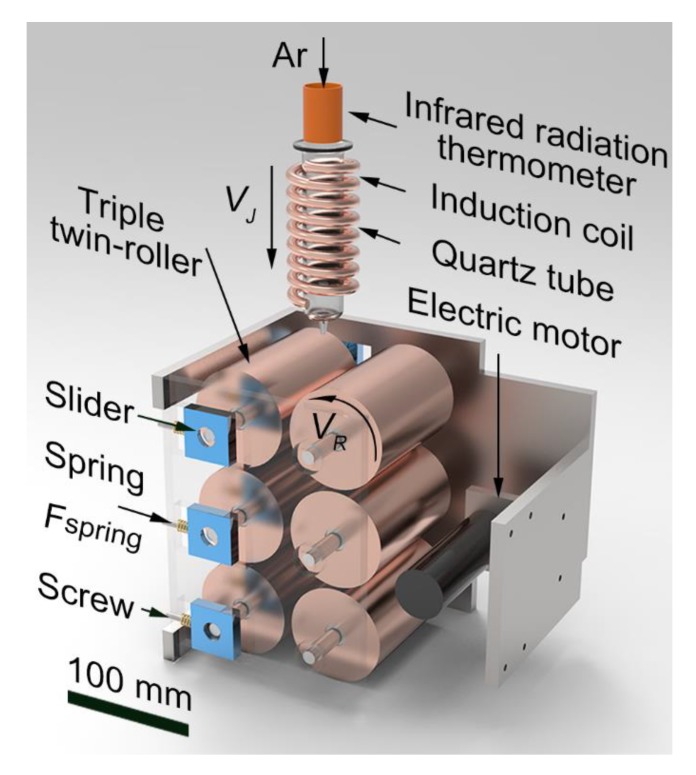
The 3D model of the triple twin-roller casting device.

**Figure 2 materials-12-03842-f002:**
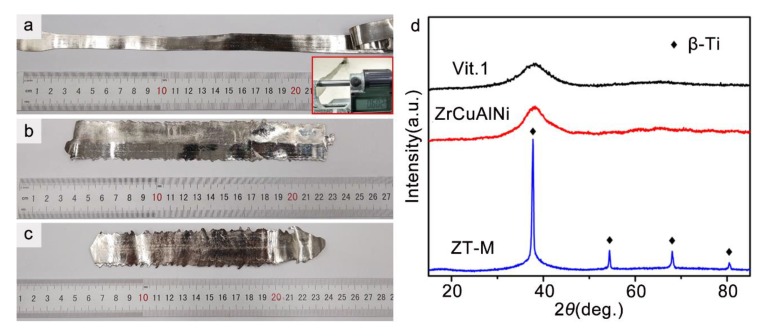
The metallic glass (MG) and metallic glass composite (MGC) strips produced via triple twin-roller casting technique, (**a**) Vit.1 MG with an inset of the measurement of its thickness, (**b**) ZrCuAlNi MG, and (**c**) ZT-M MGC strips. (**d**) The X-ray diffraction (XRD) spectra of the strips.

**Figure 3 materials-12-03842-f003:**
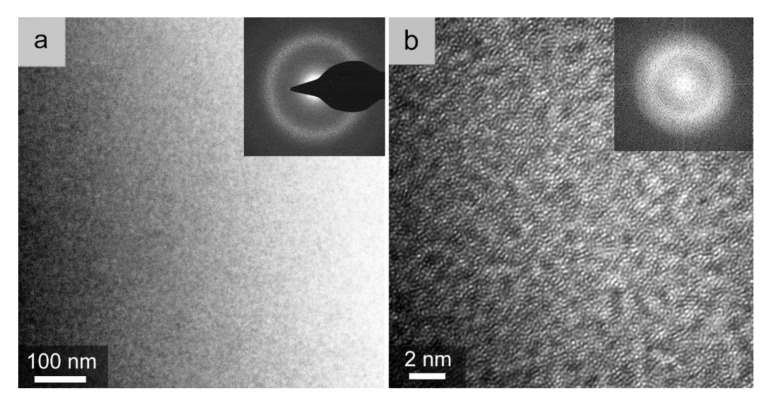
(**a**) Transmission electron microscopy (TEM) micrograph and (**b**) high-resolution TEM (HRTEM) image of the Vit.1 strip with insets of the selected-area electron diffraction (SAED) and fast-Fourier transform (FFT) patterns, respectively.

**Figure 4 materials-12-03842-f004:**
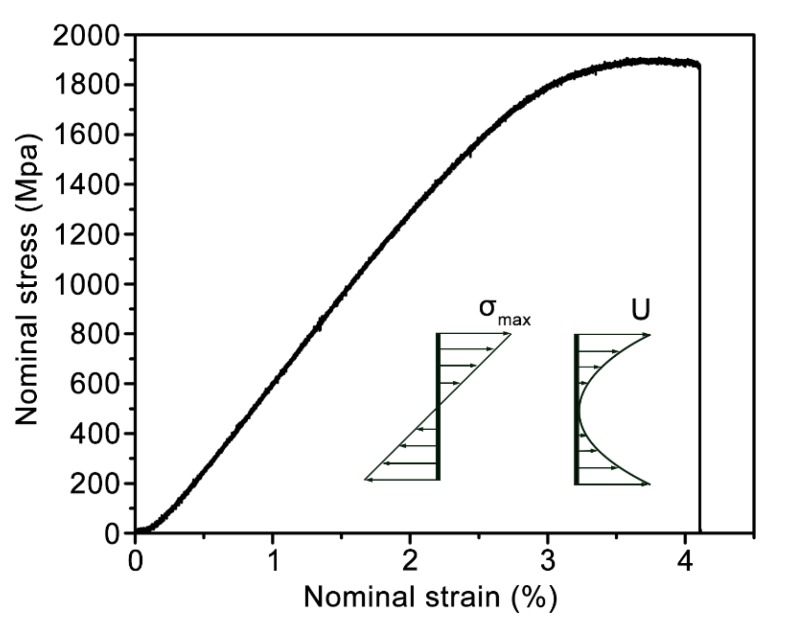
The nominal stress-strain curve of the Vit.1 strip under 3-point bending.

**Figure 5 materials-12-03842-f005:**
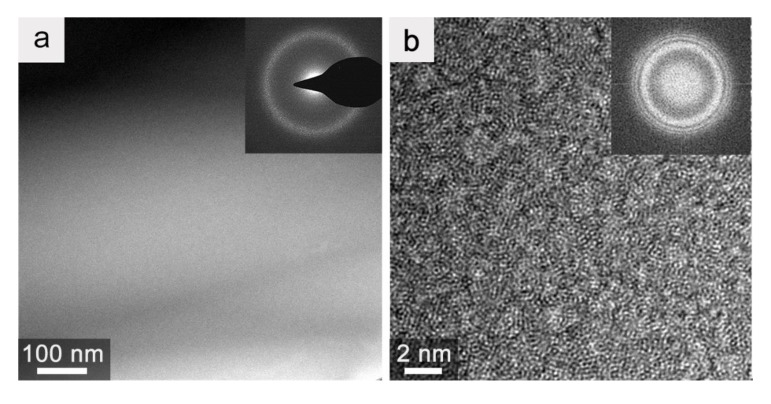
(**a**) TEM micrograph and (**b**) HRTEM image of the ZrCuAlNi strip with insets of the SAED and FFT patterns, respectively.

**Figure 6 materials-12-03842-f006:**
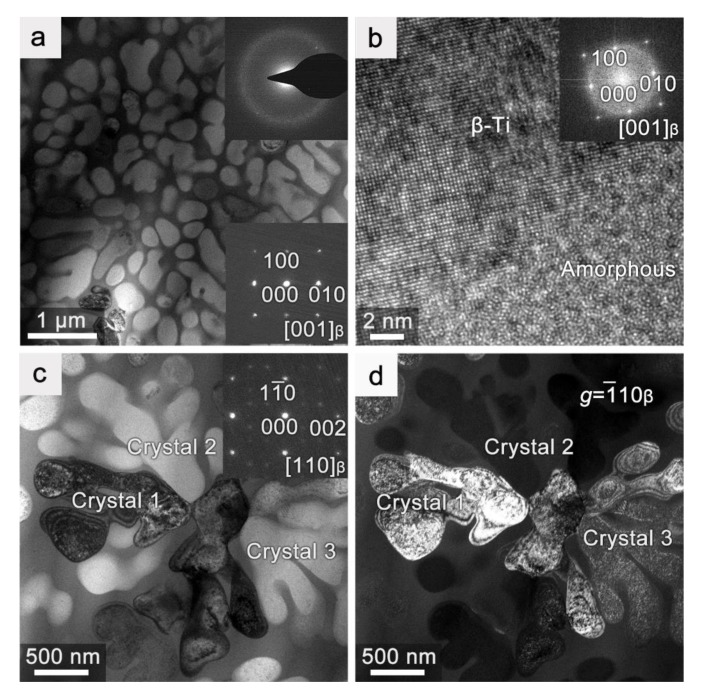
(**a**) TEM micrographs of the ZT-M strip with insets of SAED patterns from the glassy matrix (top) and from β-Ti (bottom). (**b**) HRTEM image taken from the glass/β-Ti interfacial region with the FFT pattern (inset). (**c**) and (**d**) are the bright-field and dark-field TEM micrographs, respectively, of a dendrite in the ZT-M strip.

**Table 1 materials-12-03842-t001:** The used parameters for making the MG/MGC strips and the dimensions of the prepared MG/MGC strips.

Alloy	Weight (g)	Roller Speed (r min^−1^)	Nozzle Diameter (mm)	Ar Flow Rate (L min^−1^)	Length (mm)	Width (mm)	Thickness (mm)
Vit.1	16.5	185	1	8	300	12	0.6
ZrCuAlNi	18.2	185	1.5	8	190	26	0.43
ZT-M	17.6	185	1.5	8	170	20	0.46

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
