# Peer review of "A Multiple Twin-Roller Casting Technique for Producing Metallic Glass and Metallic Glass Composite Strips"

_materials, 2019, doi:10.3390/ma12233842_

Round 1

Reviewer 1 Report

The paper presents some interesting results related to the fabrication of thick strips (> 400 um) from metallic glass and metallic glass composite alloys. The idea of using a triple twin-roller device to produce thick ribbons is innovative and deserves to be published.

However, the paper is not suitable for publication in the present form. First, it requires extensive editing of English language.

In addition, the authors should provide more information concerning the parameters used during the preparation of the thick ribbons: the rollers speed, the distance between the crucible nozzle and the first set of twin-rollers, etc.

They should also present more clearly the parameters used for each of the 3 chosen compositions and should justify their choice.

How they can explain the differences in the thicknesses and lengths of the strips prepared from the 3 different compositions.

Why is the method they are proposing more convenient comparing with the conventional casting methods?

Is the method applicable also to materials with reduced glass-forming ability?

Author Response

Please see the attached PDF file

Reviewer 2 Report

Thank you. 
